# Safety Monitoring of COVID-19 Vaccines in Persons with Prior SARS-CoV-2 Infection: A European Multi-Country Study

**DOI:** 10.3390/vaccines12030241

**Published:** 2024-02-26

**Authors:** Francesco Ciccimarra, Nicoletta Luxi, Chiara Bellitto, Luca L’Abbate, Monika Raethke, Florence van Hunsel, Thomas Lieber, Erik Mulder, Fabio Riefolo, Caroline Dureau-Pournin, Andreea Farcas, Francisco Batel Marques, Kathryn Morton, Debabrata Roy, Simona Sonderlichová, Nicolas H. Thurin, Felipe Villalobos, Miriam C. Sturkenboom, Gianluca Trifirò

**Affiliations:** 1Department of Diagnostics and Public Health, University of Verona, 37134 Verona, Italy; 2Department of Medicine, University of Verona, 37134 Verona, Italy; 3Netherlands Pharmacovigilance Centre Lareb, Goudsbloemvallei 7, 5237 MH ‘s-Hertogenbosch, The Netherlands; 4Department of PharmacoTherapy, Epidemiology & Economics, Groningen Research Institute of Pharmacy (GRIP), University of Groningen, 9712 CP Groningen, The Netherlands; 5Teamit Institute, Partnerships, Barcelona Health Hub, 08025 Barcelona, Spain; 6University of Bordeaux, INSERM CIC-P 1401, Bordeaux PharmacoEpi, 146 rue Léo Saignat, 33076 Bordeaux, France; 7Pharmacovigilance Research Center, Iuliu Hatieganu University of Medicine and Pharmacy, 400347 Cluj-Napoca, Romania; 8Laboratory of Social Pharmacy and Public Health, School of Pharmacy, University of Coimbra, 3000-548 Coimbra, Portugal; 9Drug Safety Research Unit, Southampton SO31 1AA, UK; 10University of Portsmouth, Portsmouth PO1 2UP, UK; 11Faculty of Medicine, SLOVACRIN, Pavol Jozef Šafárik University in Košice, 040 01 Košice, Slovakia; 12Fundació Institut Universitari per a la Recerca a l’Atenció Primària de Salut Jordi Gol i Gurina (IDIAPJGol), 08007 Barcelona, Spain; 13Department of Data Science and Biostatistics, Julius Global Health, University Medical Centre Utrecht, 3584 CG Utrecht, The Netherlands

**Keywords:** people with prior SARS-CoV-2 infection, Covid Vaccine Monitor, COVID-19 vaccines, adverse event, active surveillance

## Abstract

In all pivotal trials of COVID-19 vaccines, the history of previous SARS-CoV-2 infection was mentioned as one of the main exclusion criteria. In the absence of clinical trials, observational studies are the primary source for evidence generation. This study aims to describe the patient-reported adverse drug reactions (ADRs) following the first COVID-19 vaccination cycle, as well as the administration of booster doses of different vaccine brands, in people with prior SARS-CoV-2 infection, as compared to prior infection-free matched cohorts of vaccinees. A web-based prospective study was conducted collecting vaccinee-reported outcomes through electronic questionnaires from eleven European countries in the period February 2021–February 2023. A baseline questionnaire and up to six follow-up questionnaires collected data on the vaccinee’s characteristics, as well as solicited and unsolicited adverse reactions. Overall, 3886 and 902 vaccinees with prior SARS-CoV-2 infection and having received the first dose or a booster dose, respectively, were included in the analysis. After the first dose or booster dose, vaccinees with prior SARS-CoV-2 infection reported at least one ADR at a higher frequency than those matched without prior infection (3470 [89.6%] vs. 2916 [75.3%], and 614 [68.2%] vs. 546 [60.6%], respectively). On the contrary side, after the second dose, vaccinees with a history of SARS-CoV-2 infection reported at least one ADR at a lower frequency, compared to matched controls (1443 [85.0%] vs. 1543 [90.9%]). The median time to onset and the median time to recovery were similar across all doses and cohorts. The frequency of adverse reactions was higher in individuals with prior SARS-CoV-2 infection who received Vaxzevria as the first dose and Spikevax as the second and booster doses. The frequency of serious ADRs was low for all doses and cohorts. Data from this large-scale prospective study of COVID-19 vaccinees could be used to inform people as to the likelihood of adverse effects based on their history of SARS-CoV-2 infection, age, sex, and the type of vaccine administered. In line with pivotal trials, the safety profile of COVID-19 vaccines was also confirmed in people with prior SARS-CoV-2 infection.

## 1. Background

All COVID-19 vaccines authorized by the European Medicines Agency (EMA) showed a good benefit–risk profile in pivotal clinical trials [1]. The most commonly reported local and systemic adverse reactions in these studies were injection site pain, fatigue, headache, erythema, and induration; a smaller percentage of participants reported injection site redness or swelling. The incidence of serious adverse events was low, and similar in the vaccine and placebo groups [2,3]. However, vulnerable populations/higher-risk-people, and special cohorts, such as those with prior SARS-CoV-2 infection, have not been included in these trials [4].

Assessing the benefit–risk profile of COVID-19 vaccines in the real-world setting, especially in those categories not recruited in clinical trials, is crucial to ensuring that they perform as intended, rapidly identifying any potential safety signal and providing insights into whether vaccination strategies need to be adapted.

Vaccination policies for people with prior SARS-CoV-2 infection evolved due to rapidly accumulated evidence on the vaccines’ benefit–risk profiles, and also because of the growing number of subjects that have been infected over time [5,6,7]. Observational studies showed that increased time between infection and vaccination might result in improved immune responses to vaccination and a lower risk of reinfections [8]. In Europe, to administer any COVID-19 vaccine, a minimum of three months of lag time after infection has been recommended [9]. In addition, the World Health Organization (WHO) and other public health agencies recommended implementing intensive post-marketing safety monitoring for these vaccinees [10].

In the absence of data from clinical trials, observational studies are the primary source for evidence generation [11,12] about the safety of COVID-19 vaccines in patients with prior SARS-CoV-2 infection.

Recently published post-marketing studies showed that COVID-19 vaccines may induce higher immunogenicity in previously infected individuals, as documented by higher anti-spike antibody titers, as compared to those without prior SARS-CoV-2 infection [13,14,15,16].

From a clinical perspective, this translates into a higher reactogenicity of COVID-19 vaccines among individuals previously infected with SARS-CoV-2, as reported in both prospective and retrospective observational studies published so far (Appendix A).

However, most of these observational studies were conducted in single countries and assessed the safety of only one or, at most, two vaccine brands; were restricted to either the first vaccination cycle or a booster dose only; and monitored a short follow-up period. To date, no observational studies assessing the comparative safety of all COVID-19 vaccines authorized by the EMA across different vaccine brands and doses in persons with prior SARS-CoV-2 infection, as compared to SARS-CoV-2 infection-history naïve vaccinees, have been published.

The EMA funded the European multi-country project “Covid Vaccine Monitor” (CVM), a large-scale cohort event monitoring system aimed at collecting vaccinee-reported outcomes on the safety of all EMA-approved COVID-19 vaccines through web-based questionnaires within the general population, as well as in special categories, including those with prior SARS-CoV-2 infection.

As part of the CVM project, this study aims to investigate the safety profiles of different doses and brands of all EMA-licensed COVID-19 vaccines in people with previous SARS-CoV-2 infection, matched 1:1 to prior infection-free persons, from eleven European countries.

## 2. Methods

### 2.1. Setting and Study Population

This was a prospective cohort study based on electronic questionnaires collecting vaccinee-reported characteristics and outcomes from 11 European countries.

All vaccinees, who provided informed consent to participation in the study, registered on the web app within 48 h after receiving the first or a booster dose of any EMA-authorized COVID-19 vaccine (Vaxzevria, Comirnaty, Spikevax, or Jcovden vaccines) in a period ranging from February 2021 to February 2023 and reported a prior SARS-CoV-2 infection before vaccination.

Applying the same inclusion criteria, as the control group, a random sample of vaccinees participating in the CVM project and who did not report prior SARS-CoV-2 infection at the vaccination time was selected.

Controls were 1:1 matched to vaccinees with prior SARS-CoV-2 infection by gender, age, and vaccine dose and brand at the study entry. The matching procedure was performed based on the propensity score values, using the nearest-neighbor procedure for the age variable and exact matching for gender, dose, and vaccine brand.

In each country, the study protocol was approved by the respective Ethics Committee, and the study was carried out in line with the General Data Protection Regulation (GDPR) and the Data Protection Impact Assessment (DPIA). The study protocol has been submitted to the EU-PAS register (EUPAS42504).

### 2.2. Data Collection

Data were collected by partners of the CVM project, a large European network including pharmacovigilance centers, universities, hospitals, and local health units, as well as the EMA. Vaccinees were invited to participate in the cohort event-monitoring study through ad hoc dissemination materials (e.g., flyers, posters, animation videos, and infographics), which were distributed through different channels, such as print magazines, online journals, scientific society web pages, social networks, and vaccination centers.

Two specific and harmonized web-based apps were developed for data collection, the Lareb-managed Intensive Monitoring (LIM) and Research Online (RO), which were built specifically for vaccinee-reported outcomes, and questionnaires were translated into local languages [17,18].

In the baseline questionnaire, information on vaccinees’ characteristics, including demographics, medical history (i.e., cardiovascular disorder, diabetes mellitus, hypertension, liver disorder, lung disorder, mental disorder, malignant tumor, nervous system disorder, and renal disorder), concomitant drug use, and administered COVID-19 vaccine dose and brand, were collected. In addition, for people with prior SARS-CoV-2 infection, information on the dates and symptoms relevant to the infection was collected. Moreover, vaccinees filled out a total of six follow-up questionnaires (FU-Qs) during the six months after the first dose of COVID-19 vaccines, and five follow-up questionnaires over a 3-month follow-up period were provided for those recruited at the booster dose (Figure 1). Participants could register for the booster dose via the web app until 30 November 2022. FU-Qs were stored until 28 February 2023.

FU-Qs collected information on patient-reported short-/medium-/long-term adverse reactions experienced following the COVID-19 vaccines. Specifically, FU-Qs collected data on solicited local reactions (injection site hematoma, induration, inflammation, pain, pruritus, swelling, and warmth) and systemic adverse drug reactions (ADRs) (arthralgia, chills, fatigue, headache, malaise, myalgia, nausea, and fever), based on the most frequently reported adverse events in pivotal trials [2,3]. Information on unsolicited ADRs, adverse events of special interest (AESI), and serious ADRs was also collected. Serious ADRs were assessed and coded by pharmacovigilance-trained personnel according to the Council for International Organizations of Medical Sciences (CIOMS criteria) [19]. Adverse events of special interest were defined according to a list provided by the Brighton Collaboration [20]. All patient-reported adverse reactions were coded using Medical Dictionary for Regulatory Activities (MedDRA) terminology 23.0 and 24.0 [21] and ultimately sent to EudraVigilance. For each ADR, time of onset, outcome, duration of symptoms (if recovered), and severity/impact of symptoms (including medical assistance and hospitalization) were topics of inquiry [22].

### 2.3. Data Analysis

Data from 11 European countries (Belgium, France, Italy, Ireland, Portugal, Romania, Slovakia, Spain, Switzerland, the Netherlands, and the United Kingdom), were collected through two comparable tools and standardized using a common data model (CDM) and then pooled and analyzed centrally. Only vaccinees who filled out the baseline and at least the first follow-up questionnaire were included in the analyses.

First, a descriptive analysis of the demographics and clinical characteristics of people with prior SARS-CoV-2 infection versus the matched control was conducted and stratified by first/booster dose as well as vaccine brands.

Second, the proportion of subjects with at least one ADR in general, as well as specifically solicited/unsolicited/serious ADRs, was also evaluated, using as denominator the total number of recruited vaccinees receiving the first dose, second dose, or a booster dose of any vaccine brand, and who filled in (baseline plus) at least the first FU-Q. In addition, the frequency of local and systemic solicited ADRs reported after the first, second, or booster doses of any vaccine, for people with prior SARS-CoV-2 infection versus matched controls, was measured. To investigate a potential interaction effect of medical history on the association between previous SARS-CoV-2 infection and the occurrence of at least one ADR, logistic regression models were utilized. The estimated interaction effect was represented in a forest plot. In addition, to explore the occurrence of ADRs in relation to SARS-CoV-2 variants of concern, participants who reported the date of symptom onset in the web app were divided into two groups based on date periods. Based on a previous study [23], we estimated the prevalence of SARS-CoV-2 variants using the WHO periodic bulletin, identifying two periods (i.e., Alpha and Delta). The occurrence of ADRs in relation to the severity of COVID-19 symptoms reported in the baseline questionnaire (i.e., no symptoms, symptoms similar to cold, many symptoms, or admitted to hospital) was also analyzed.

Third, heatmaps of the proportions of participants who reported at least one solicited ADR, stratified by gender and age categories, were generated. In addition, heatmaps of the percentages of vaccinees who reported local and systemic solicited ADRs, stratified by vaccine brand, dose, gender, and age categories, were also produced.

Fourth, the time to onset (TTO) and the time to recovery (TTR) of reported ADRs were analyzed and visualized with a combination of violin plots and box-plots, with median, first quartile, third quartile, minimum, and maximum observed values rendered in hours. Participants could report the TTO and TTR of an ADR as the number of seconds, minutes, hours, days, weeks, or months, or with a specific calendar date. For this analysis, only subjects who reported both TTO and TTR for a specific ADR were included.

All of the heatmaps and TTO and TTR analyses for vaccinees with prior SARS-CoV-2 infections versus matched controls were separately generated. Finally, we explored whether the frequency of ADR reporting among subjects with prior SARS-CoV-2 infection varied based on the time elapsed between infection and vaccination, using four different timeframes (i.e., 0–90 days, 91–180 days, 181–360 days, and >360 days).

Categorical variables were reported as absolute frequencies and percentages, while continuous variables were reported as median (interquartile range), as appropriate. A Chi-square test or Fisher’s exact test was performed to compare categorical variables, while a Wilcoxon test was performed to compare medians, as appropriate. A *p*-value < 0.05 denoted statistical significance. All analyses were performed using R statistical software (version 4.3.1). For heatmaps, violin plots, box-plots, and random forest analysis, the ggplot2 R package was used.

## 3. Results

Overall, between February 2021 and November 2022, 37,170 subjects from 11 European countries registered for the Covid Vaccine Monitor study and completed both the baseline questionnaire and the FU-Q1. Of them, 30,186 (81.2%) registered after receiving the first vaccination cycle and 6984 (18.9%) after receiving the booster dose.

From this study population, overall, 12.9% of vaccinees recruited at the first vaccination cycle (N = 3886) and at the booster dose (N = 902) reported a prior SARS-CoV-2 infection at the time of vaccination. These vaccinees were included in the analyses and 1:1 matched to a cohort of vaccinees without a prior SARS-CoV-2 infection by age, gender, and vaccine brand.

Regarding subjects included in the analyses, after the first vaccination cycle, 50.8% of the vaccinees with prior SARS-CoV-2 infection and 56.6% of the matched controls filled out all six FU-Qs, while only 41.2% of the vaccinees with prior SARS-CoV-2 infection and 50.6% of the matched controls completed all five FU-Qs after the booster dose (Figure 2).

Most of the participants with prior SARS-CoV-2 infection received Vaxzevria (N = 1410; 36.3%) at the first vaccination cycle, a percentage followed by Comirnaty (N = 1372; 35.6%), Spikevax (N = 668; 17.2%), and Janssen (N = 422; 10.9%). The female/male (F/M) ratio was equal to 2.8 (Table 1).

As for the booster dose, most of the participants (F/M ratio = 1.9) received Comirnaty (N = 521; 57.8%), a percentage followed by Spikevax (N = 376; 41.2%), while only three received Vaxzevria (0.3%) (Table 2). The median ages of participants who received the first and the booster doses were 46 years (interquartile range: 33–56 years) and 42 years (interquartile range: 31–53 years), respectively. In general, no statistically significant differences in terms of medication use and medical history were observed among subjects with prior SARS-CoV-2 infection and matched controls, for both the first vaccination cycle and booster dose (Table 1 and Table 2).

After the first and the booster dose, vaccinees with prior SARS-CoV-2 infection reported at least one ADR with higher frequencies than those matched without prior infection (3470 [89.6%] vs. 2916 [75.3%], and 614 [68.2%] vs. 546 [60.6%], respectively). On the contrary side, after the second dose, vaccinees with a history of SARS-CoV-2 infection reported at least one ADR with a lower frequency compared to matched controls (1443 [85.0%] vs. 1543 [90.9%]) (Figure 3, Appendix A).

In more detail, the proportion of vaccinees with prior SARS-CoV-2 infection reporting at least one ADR was higher for those who received Vaxzevria for the first dose and for those who received Spikevax for the second or booster doses (Appendix A). As shown in Appendix A, for the first dose and the booster dose, no statistically significant interaction effect of comorbidities on the association between previous SARS-CoV-2 infection and the occurrence of ADRs was observed, except for hypertension at the first dose. However, for both the first and the booster dose, a trend of increased risk for diabetes was observed, although it does not reach the point of significance (Appendix A).

In both cohorts, irrespective of vaccine dose and brand, injection site pain was the most frequently reported local solicited ADR, while fatigue, headache, malaise, and myalgia were the most reported systemic solicited ADRs (Figure 4).

Regarding the booster dose, arthralgia, chills, malaise, increased body temperature, and pyrexia were significantly more reported among subjects with prior SARS-CoV-2 infection who received Comirnaty as compared with their matched controls (*p* < 0.05 for each comparison) (Appendix A).

Among all doses and brands of vaccines, the frequency of AESIs was low, ranging from 0.1% to 1.2%. A similar trend was also observed for serious ADRs, with frequencies ranging from 0.1% to 0.3% (Appendix A). The total numbers of reported serious ADR and AESI (n = 85 in total), stratified by vaccine brand, dose, and cohort, are shown in Table 3 and Table 4.

When stratifying by different timeframes between SARS-CoV-2 infection and vaccine administration, no significant differences in the frequency of reported ADRs were observed (Appendix A). However, a statistically significant difference was observed in the frequency of reported ADRs by stratifying the analysis by variants of concern following the first dose (*p*-value = 0.006) (Appendix A).

The frequency of people with prior SARS-CoV-2 infection reporting at least one ADR, stratified by different severity of symptoms, showed a statistically significant difference for both first and booster doses (*p*-value < 0.001) (Appendix A).

Overall, across all doses, women and younger subjects were more likely to report at least one solicited ADR, despite no sex-related differences being found in the distribution of systemic ADRs observed (Appendix A, and Figure 5).

As shown in the violin plots in Figure 6A, the median time to onset for all reported ADRs following the first dose was 12.9 (interquartile range: 7.4–22.7) hours among vaccinees with prior SARS-CoV-2 infection and 11.7 (interquartile range: 6.2–23.1) hours among the matched controls. Likewise, a similar time to recovery in people with prior SARS-CoV-2 infection (37.9 h; interquartile range 21.3–64.3 h) versus matched controls (37.3 h; interquartile range 19.8–57.8 h) was reported (Figure 6B).

Following the second dose, a similar time to onset was observed among infected and matched controls (Figure 6C), whilst the time to recovery was higher for matched controls (37.2 h; interquartile range 20.1–59.3), as compared with participants with previous SARS-CoV-2 infection (28.3 h; interquartile range 19.8–53.5) (Figure 6D).

Lastly, regarding the booster dose, people with prior SARS-CoV-2 infection showed a slightly lower median TTO (13.5 h; interquartile 1.8–20.0), than that of the matched controls (15.5 h; interquartile 2.9–20.9) (Figure 6E). Time to recovery demonstrated the same trend as that observed for the second dose (Figure 6F). A statistically significant difference for the TTO and TTR of individuals with prior SARS-CoV-2 infection and participants without prior infection in each dose was observed, except for that for the TTO of individuals who received the second dose.

## 4. Discussion

To our knowledge, this is the first active surveillance study that explored the safety of COVID-19 vaccines through patient-reported outcomes, collected via electronic questionnaires, in people with prior SARS-CoV-2 infection, as compared to a matched cohort of not previously infected vaccinees, across different vaccine doses and brands from multiple European countries, with a long study period. In addition, as compared to previously published observational studies that focused exclusively on single vaccine brands or doses, our evidence provided a broader overview of different COVID-19 vaccine brands and doses authorized by the EMA.

This study documented that the safety profile of COVID-19 vaccines in people with prior SARS-CoV-2 infection is favorable, with extremely low frequencies of serious adverse reactions/AESI following each vaccine dose. Overall, subjects with prior SARS-CoV-2 infection were more likely to experience at least one ADR, as compared to those without a history of SARS-CoV-2 infection, after the first dose or the booster dose, confirming already-existing evidence coming from several observational studies [24,25]. In particular, for both cohorts, the most frequently reported local solicited ADR following both the first vaccination cycle and the booster dose was injection site pain. As for systemic solicited ADRs, fatigue, headache, malaise, and myalgia were the most commonly reported; such findings are in line with previously published articles [26,27,28].

Furthermore, as already reported in the scientific literature, our study demonstrated that, for all vaccine brands, the frequency of ADRs among individuals with prior SARS-CoV-2 infection was higher following the first dose than the second dose [24,25]. This finding is also confirmed by a longitudinal study conducted on health workers in the Johns Hopkins Health System, which reported that people with previous SARS-CoV-2 infection were associated with an increased risk of experiencing clinically significant symptoms following a first dose of Spikevax or Comirnaty vaccines, as compared to the second dose [29]. It has been shown that COVID-19 vaccines have increased immunogenicity in individuals with past infection due to higher antibody titers than found in those without previous infection [30,31,32]. In particular, Krammer et al. documented that individuals with pre-existing immunity have relevant spike antibody responses and experience more severe reactogenicity after the first dose, as compared to naïve individuals [33]. In addition, no increase in antibody titers in individuals with a history of COVID-19 was observed following the second vaccine dose, thus potentially explaining the lower frequency of ADRs reported [33]. Moreover, a U.K. prospective cohort study collecting information through an ad hoc developed app on self-reported ADRs following COVID-19 vaccination reported higher reactogenicity after the first dose of the Vaxzevria vaccine [34]. Additionally, an active-surveillance cohort study conducted in Canada found a greater reactogenicity among subjects with prior SARS-CoV-2 infection following the administration of both the second and the booster dose of Spikevax vaccine, as compared to Comirnaty and Vaxzevria [35]. In a longitudinal study, Debes et al. observed that Spike IgG antibody measurements were higher in individuals who received the Spikevax vaccine, had prior SARS-CoV-2 infection, and reported clinically significant reactions [29]. Such evidence supports our findings concerning the vaccine brands associated with a higher frequency of ADRs, as reported following both the first vaccination cycle and the booster dose. Overall, based on these observations, the association between more severe symptomatology and a higher reported frequency of ADRs could be related to an increase in antibody titers. Furthermore, the variant analysis of SARS-CoV-2 also suggests a likely change in antibody titer and, consequently, a variation in reported ADR frequencies. As expected, most ADRs were reported by women and younger subjects, in both study cohorts. Specific sex-related immunological, hormonal, and genetic differences in immune responses are widely documented in the scientific literature [36], and it is also well-known that antibody titers tend to decrease as age increases [37]. The lower frequency of ADRs reported after the booster dose, as compared to the first vaccination cycle, may be related to the lower number of follow-up questionnaires scheduled for this dose. Consequently, this might have allowed a lower frequency of ADRs to be captured. However, as reported in previous studies [20,31], and in line with our results, the majority of reported ADRs occur within 8 days of vaccination. In addition, following the first vaccination cycle, there has been a growing awareness of adverse events associated with the administration of COVID-19 vaccines, which could have made people less motivated to participate in the study or report ADRs. The high frequency of ADRs reported following the first vaccination cycle, as compared to the booster dose, could be explained by the heterogeneity of the vaccines administered during the first vaccination cycle. In particular, although Vaxzevria showed a high frequency of ADRs reported after the first and second doses, it was excluded from the analysis of the booster dose, since only three participants received this vaccine brand. Consequently, the overall lower frequency of ADRs reported after the booster dose could be related to the absence of data related to this vaccine brand. Furthermore, our study shows that people with previous SARS-CoV-2 infection and a history of hypertension are more likely to report at least one ADR than those without a history of hypertension. One of the main strengths of this study is that it included patient-level data from eleven European countries, which were collected and analyzed using a CDM. Moreover, the adaptability of the LIM and RO web apps in integrating information that became available after the study started and in adapting to changing regulatory frameworks (e.g., variations in vaccination regimens) should also be acknowledged. It was possible to directly and promptly update the questionnaires by adding new questions or modifying questions to capture data to test emerging hypotheses about COVID-19-related signs and symptoms. In addition, the large sample number of vaccinees enrolled allowed us to investigate vaccine safety and to compare safety outcomes among subjects with and without prior SARS-CoV-2 infection. Another strength of our study is the duration of the study period as well as the follow-up period. Previously published studies examining the safety of the COVID-19 vaccine in persons with previous SARS-CoV-2 infection were conducted for a period ranging from 1 month to 12 months, and their follow-up periods ranged from 7 days to 4 months. Only one study with a 12-month follow-up period was identified (Appendix A). Lastly, unlike passive surveillance studies conducted using spontaneous reporting system databases, the availability of a denominator allowed for the assessment of the frequency of specific ADRs as well as the stratification and adjustment of the analyses. However, some limitations warrant caution. First, particularly serious ADRs, e.g., those leading to hospitalization or severe disability, might have been missed if vaccinees experiencing such ADRs were unable to use the web-based apps to report them. As a consequence, the frequency of serious ADRs could have been underestimated. However, a study on surveillance of adverse events following immunization shows that self-reported data described more serious events than those reported by healthcare professionals [38]. In addition, participants whose reminder emails for the completion of the follow-up questionnaire ended up in spam or who did not experience any ADRs may have contributed to the loss to follow-up. Furthermore, after the first vaccination cycle, there was an increasing consciousness of the safety of these vaccines, which may have made vaccinees less motivated to continue filling out the FU-Qs, contributing to the loss of follow-up data. Second, considering that self-reporting of health-related events may be subject to recall bias, people with previous SARS-CoV-2 infections may be more likely to report them. Such bias may concern both symptomatic (especially if they have had a more severe infection) and asymptomatic patients. The latter may indeed have been worried about their own health, and thus tending to recall events more precisely. Third, it should be considered that there may be periods of overlap between variants of concern, and this could affect the analysis, as stratified by variant type. In addition, since information on ethnicity was not recorded, our findings might not be entirely applicable to certain groups. Fourth, it should be taken into account that self-reported information is subjective in nature and participants may have misinterpreted or misreported information. However, the coding of ADRs and their seriousness was assessed by trained pharmacovigilance personnel who could contact the participant if further information were needed, in cases in which the participant agreed during registration on the web app to be recontacted. In addition, the prior SARS-CoV-2 infection was not confirmed by a clinician, although the majority of included vaccinees reported a diagnosis of SARS-CoV-2 infection confirmed with a positive polymerase chain reaction (PCR) test. Lastly, the individuals who utilized the web app were self-selected and may not adequately represent the general population.

## 5. Conclusions

This large-scale prospective study of COVID-19 vaccinees allowed the comparison between different brands and doses of vaccines in people previously infected with SARS-CoV-2 and individuals without a history of infection, by administering the same survey to all participants in eleven European countries. Adverse reactions were reported more frequently by those with a previous SARS-CoV-2 infection than those without a history of SARS-CoV-2 infection; this was observed in both cohorts. The frequency of reported ADRs was higher in young subjects and women. The frequency of serious ADRs was low for all doses and cohorts. Overall, post-vaccinal symptoms (both systemic and local) resolved within a few days after administration of the first, second, or booster doses.

## Figures and Tables

**Figure 1 vaccines-12-00241-f001:**
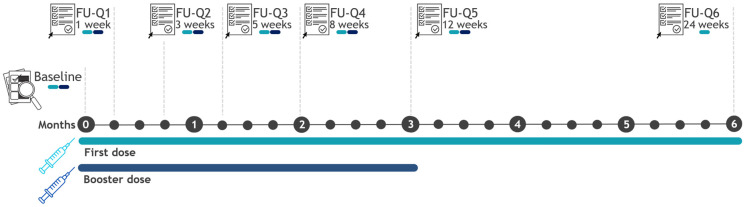
Questionnaire scheduling scheme over time. Abbreviations: FU-Q = Follow-up questionnaire.

**Figure 2 vaccines-12-00241-f002:**
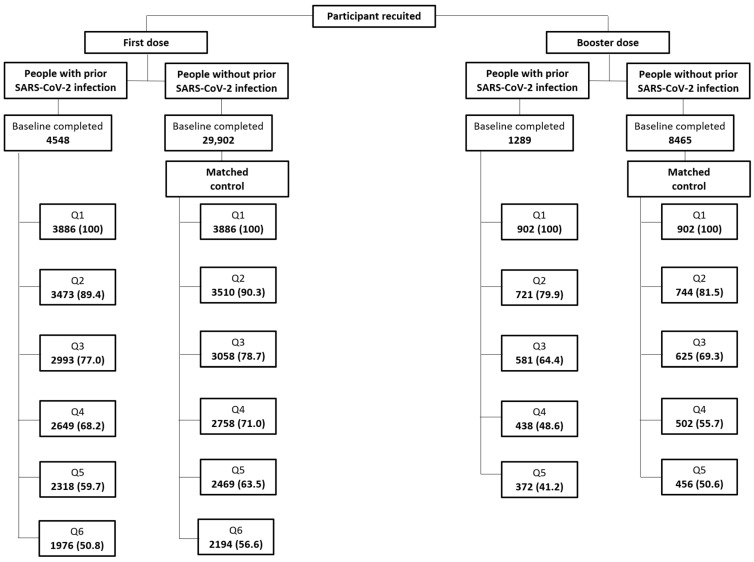
Flowchart of recruited people with prior SARS-CoV-2 infection and matched control who were recruited at either first or booster dose and who filled out different questionnaires.

**Figure 3 vaccines-12-00241-f003:**
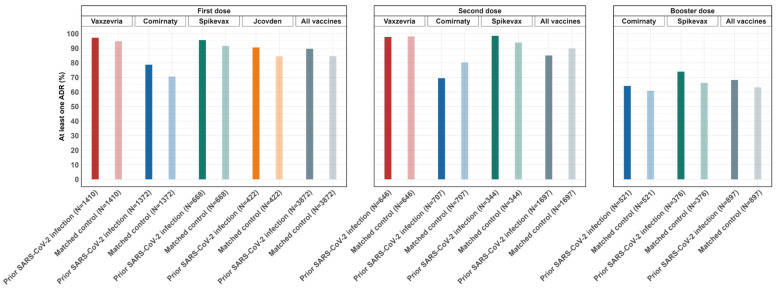
Frequency of at least one ADR reported, stratified by vaccine brand and dose, for persons with previous SARS-CoV-2 infection and matched controls.

**Figure 4 vaccines-12-00241-f004:**
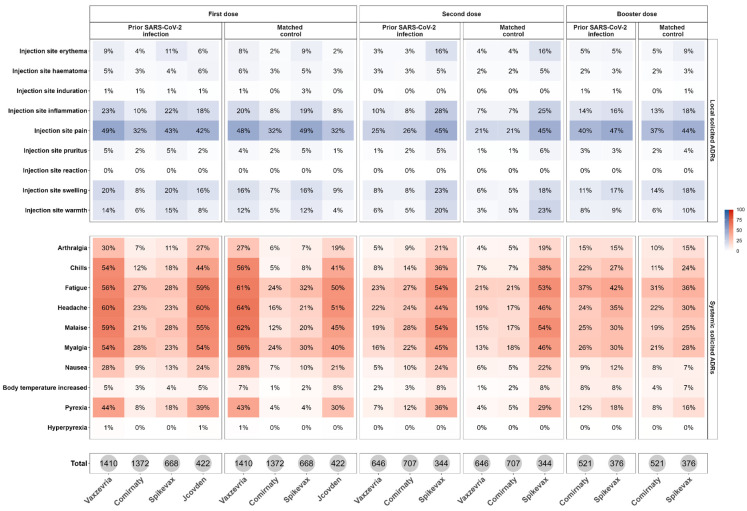
Heatmaps of the frequency of vaccinee-reported local and systemic solicited ADRs following the first, second, or booster doses, stratified by vaccine brands, for people with prior SARS-CoV-2 infection and the matched control.

**Figure 5 vaccines-12-00241-f005:**
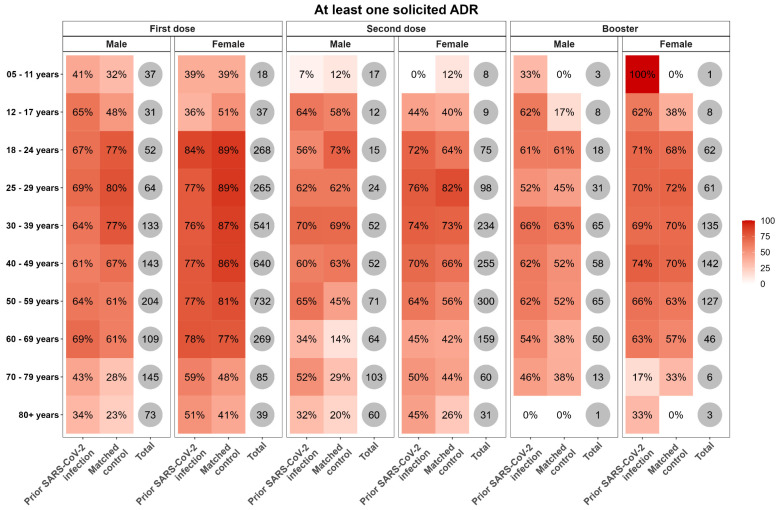
Heatmaps of the frequency of vaccinee-reported solicited ADRs following the first, second, or booster doses of any vaccine, for people with SARS-CoV-2 infection and the matched control, stratified by age categories.

**Figure 6 vaccines-12-00241-f006:**
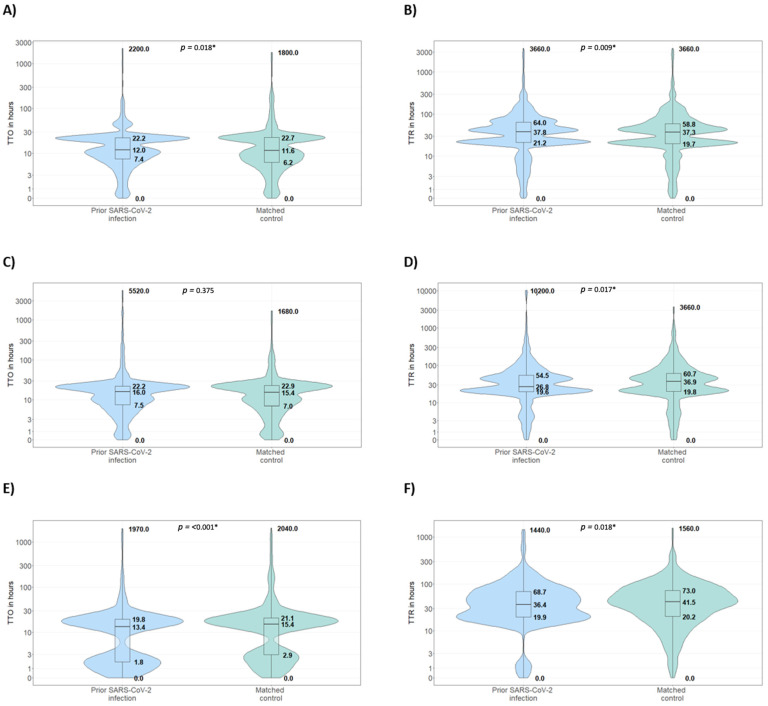
Combination of violin plots and box-plots of the median time to onset and the median time to recovery (in hours) of vaccinee-reported ADRs following a first vaccination cycle or a booster dose, in people with prior SARS-CoV-2 infection vs. matched controls. (**A**) Median time to ADR onset (in hours) following the first dose. (**B**) Median time to recovery (in hours) following the first dose. (**C**) Median time to ADR onset (in hours) following the second dose. (**D**) Median time to recovery (in hours) following the second dose. (**E**) Median time to ADR onset (in hours) following the booster dose (**F**) Median time to recovery (in hours) following the booster dose. Abbreviations: * = statistically significant.

**Table 1 vaccines-12-00241-t001:** Demographic and clinical characteristics of people with prior SARS-CoV-2 infection and the matched control, following the administration of the first COVID-19 vaccination cycle, stratified by vaccine brand.

	First Vaccination Cycle											
	Vaxzevria (ChAdOx1-S) N = 1410 (%)			Comirnaty (BioNTech/Pfizer) N = 1372 (%)			Spikevax (Moderna) N = 668 (%)			Jcovden (Ad26.COV2-S) N = 422 (%)		
	Prior SARS-CoV-2 Infection	Matched Control	*p*-Value	Prior SARS-CoV-2 Infection	Matched Control	*p*-Value	Prior SARS-CoV-2 Infection	Matched Control	*p*-Value	Prior SARS-CoV-2 Infection	Matched Control	*p*-Value
Gender												
Female	1230 (87.2)	1230 (87.2)	*Matching factor*	840 (61.2)	840 (61.2)	*Matching factor*	490 (73.4)	490 (73.4)	*Matching factor*	326 (77.3)	326 (77.3)	*Matching factor*
Male	180 (12.8)	180 (12.8)		532 (38.8)	532 (38.8)		178 (26.6)	178 (26.6)		96 (22.7)	96 (22.7)	
Median, years (IQR)	47 (35–57)	47 (35–57)		45 (31–56)	45 (31–56)		43 (32–51)	43 (32–51)		48 (36–48)	48 (36–48)	
Age Categories												
5–17	2 (0.1)	1 (0.1)	*Matching factor*	109 (7.9)	109 (7.9)	*Matching factor*	2 (0.3)	2 (0.3)	*Matching factor*	0 (0.0)	0 (0.0)	*Matching factor*
18–39	463 (32.8)	464 (32.9)		482 (35.2)	482 (35.2)		253 (37.9)	253 (37.9)		127 (30.1)	127 (30.1)	
40–69	945 (67.1)	945 (67.0)		442 (32.2)	442 (32.2)		411 (61.5)	411 (61.5)		294 (69.7)	294 (69.7)	
≥70	0 (0.0)	0 (0.0)		339 (24.7)	339 (24.7)		2 (0.3)	2 (0.3)		1 (0.2)	1 (0.2)	
Medication Use												
Use of any medication	684 (48.5)	689 (48.9)	0.131	651 (47.4)	664 (48.4)	0.647	308 (46.1)	306 (45.8)	0.956	157 (37.2)	169 (40)	0.437
Use of painkillers/fever reducing medicines ^#^	347 (24.6)	306 (21.7)	0.0742	174 (12.7)	124 (9.0)	0.00264 *	114 (17.1)	95 (14.2)	0.175	112 (26.5)	96 (22.7)	0.231
Medical History												
Cardiovascular diseases	43 (3)	50 (3.5)	0.527	110 (8)	89 (6.5)	0.141	14 (2.1)	20 (3.0)	0.385	8 (1.9)	5 (1.2)	0.576
Diabetes	32 (2.3)	38 (2.7)	0.545	50 (3.6)	50 (3.6)	1.000	11 (1.6)	9 (1.3)	0.822	1 (0.2)	1 (0.2)	1.000 ^
Hypertension	134 (9.5)	120 (8.5)	0.392	164 (12)	165 (12)	1.000	35 (5.2)	43 (6.4)	0.414	27 (6.4)	26 (6.2)	1.000
Liver diseases	32 (2.3)	29 (2.1)	0.796	23 (1.7)	22 (1.6)	1.000	9 (1.3)	23 (3.4)	0.02 *	6 (1.4)	2 (0.5)	0.287 ^
Lung diseases	2 (0.1)	3 (0.2)	1.000 ^	5 (0.4)	3 (0.2)	0.723 ^	3 (0.4)	-	0.248 ^	1 (0.2)	1 (0.2)	1.000 ^
Psychological disorders	124 (8.8)	113 (8)	0.497	109 (7.9)	97 (7.1)	0.426	70 (10.5)	70 (10.5)	1.000	18 (4.3)	22 (5.2)	0.627
Malignant tumors	50 (3.5)	60 (4.3)	0.381	50 (3.6)	69 (5)	0.0916	30 (4.5)	29 (4.3)	1.000	23 (5.5)	22 (5.2)	1.000
Neurological diseases	5 (0.4)	13 (0.9)	0.458 ^	14 (1)	15 (1.1)	1.000	4 (0.6)	5 (0.7)	1.000 ^	3 (0.7)	7 (1.7)	0.340 ^
Kidney diseases	14 (1.0)	13 (0.9)	1.000	15 (1.1)	18 (1.3)	0.726	5 (0.7)	6 (0.9)	1.000 ^	5 (1.2)	2 (0.5)	0.448 ^
Immunosuppression	10 (0.7)	8 (0.6)	0.813	14 (1)	13 (0.9)	1.000	5 (0.7)	3 (0.4)	1.000 ^	-	1 (0.2)	1.000 ^

^#^ Taken within a few hours before COVID-19 vaccine administration. Abbreviations: MedDRA = Medical Dictionary for Regulatory Activities; PT = preferred term; * = statistically significant; ^ = Fisher’s exact test. A total of 13 vaccinees who reported an unknown vaccine brand and 1 vaccinee reporting Novavax were excluded.

**Table 2 vaccines-12-00241-t002:** Demographic and clinical characteristics of people with prior SARS-CoV-2 infection and the matched control, following the administration of COVID-19 vaccine booster dose, stratified by vaccine brand.

	Booster Dose					
	Comirnaty (BioNTech/Pfizer) N = 521 (%)			Spikevax (Moderna) N = 376 (%)		
	Prior SARS-CoV-2 Infection	Matched Control	*p*-Value	Prior SARS-CoV-2 Infection	Matched Control	*p*-Value
Gender						
Female	347 (66.6)	347 (66.6)	*Matching factor*	243 (64.6)	243 (64.6)	*Matching factor*
Male	174 (33.4)	174 (33.4)		133 (35.4)	133 (35.4)	
Median, years (IQR)	41 (28–53)	41 (28–53)		43 (35–54)	43 (35–54)	
Age Categories, Years						
5–17	17 (3.3)	17 (3.3)	*Matching factor*	1 (0.3)	1 (0.3)	*Matching factor*
18–39	231 (44.3)	231 (44.3)		140 (37.2)	140 (37.2)	
40–69	259 (49.7)	259 (49.7)		226 (60.1)	226 (60.1)	
≥70	14 (2.7)	14 (2.7)		9 (2.4)	9 (2.4)	
Medication Use						
Any medication	226 (43.4)	202 (38.8)	0.148	158 (42)	130 (34.6)	0.428 *
Use of painkillers/fever reducing medicines ^#^	82 (15.7)	171 (32.8)	<0.0001 *	89 (23.7)	84 (22.3)	0.729
Medical History (MedDRA PT)						
Cardiovascular diseases	15 (2.9)	10 (1.9)	0.418	8 (2.1)	7 (1.9)	1.000
Diabetes	8 (1.5)	11 (2.1)	0.674	10 (2.7)	10 (2.7)	1.000
Hypertension	35 (6.7)	42 (8.1)	0.477	28 (7.4)	25 (6.6)	0.776
Liver diseases	17 (3.3)	13 (2.5)	0.578	5 (1.3)	4 (1.1)	1.000 ^
Lung diseases	1 (0.2)	1 (0.2)	1.000 ^	1 (0.3)	1 (0.3)	1.000 ^
Psychological disorders	37 (7.1)	33 (6.3)	0.806	26 (6.9)	14 (3.7)	0.0739
Malignant tumors	19 (3.6)	24 (4.6)	0.533	16 (4.3)	11 (2.9)	0.433
Neurological diseases	6 (1.2)	5 (1.0)	1.000 ^	2 (0.5)	1 (0.3)	1.000 ^
Kidney diseases	5 (1)	4 (0.8)	1.000 ^	2 (0.5)	2 (0.5)	1.000 ^
Immunosuppression	3 (0.6)	5 (1.0)	0.723 ^	1 (0.3)	2 (0.5)	1.000 ^

^#^ Taken within a few hours before COVID-19 vaccine administration. Abbreviations: MedDRA = Medical Dictionary for Regulatory Activities; PT = preferred term, * = statistically significant; ^ = Fisher’s exact test. A total of 2 vaccinees who reported an unknown vaccine brand and 3 vaccinees reporting Vaxzevria were excluded.

**Table 3 vaccines-12-00241-t003:** List of AESIs following the first, second, or booster doses of any vaccine.

Dose	Cohort	Vaccine Brand	AESI	N.
First dose	Matched controls	Vaxzevria	Hypersensitivity	1
			Hypersomnia	1
			Acute myocardial infarction	1
		Spikevax	Pancreatitis acute	1
		Comirnaty	Pericarditis	1
			Anaphylactoid reaction	1
			Hypersensitivity	1
	Prior SARS-CoV-2 infection	Vaxzevria	Hypersensitivity	1
		Comirnaty	Hypersensitivity	1
			Hypersensitivity	1
		Moderna	Hypersomnia	1
Second dose	Matched controls	Comirnaty	Seizure	1
		Vaxzevria	Epilepsy	1
			Myocardial infarction	1
	Prior SARS-CoV-2 infection	Vaxzevria	Hypersensitivity	1
Booster dose	Prior SARS-CoV-2 infection	Spikevax	Hypersensitivity	1

Abbreviations: AESI = adverse events of special interest; N. = number.

**Table 4 vaccines-12-00241-t004:** List of serious ADRs following the first, second, or booster doses of any vaccine.

Dose	Cohort	Vaccine Brand	Serious ADR	N.	CIOMS Criteria
First dose	Matched controls	Spikevax	Arthralgia	1	Other
			Headache	1	Other
			Myalgia	1	Other
			Paresthesia	1	Other
			Pyrexia	1	Other
			Abortion, spontaneous	2	Other
			Body temperature increased	1	Other
		Vaxzevria	Atrial fibrillation	1	Other
			Pyrexia	1	Other
			Acute myocardial infarction	1	Hospital
			Myocardial infarction	1	Hospital
		Comirnaty	Urticaria	1	Other
			Diarrhea	1	Hospital
			Hyperpyrexia	1	Hospital
			Vomiting	1	Hospital
		Jcovden	Dyspnea	1	Other
			Hypoesthesia	1	Threatening
			Limb discomfort	1	Other
			Pallor	1	Other
			Palpitations	1	Other
			Paresthesia, oral	1	Other
			Restlessness	1	Other
			Tremor	1	Other
	Prior SARS-CoV-2 infection	Vaxzevria	Pulmonary pain	1	Other
			Retinal detachment	1	Other
			Vitreous floaters	1	Other
		Comirnaty	Abortion, spontaneous	1	Other
			Aggravated condition	1	Other
			Hypersensitivity	1	Threatening
			Muscle spasms	1	Threatening
			Depression	1	Other
			Dyspnea	1	Other
			Eye hemorrhage	1	Other
			Fatigue	1	Other
			Headache	1	Other
			Hypersensitivity	1	Other
			Nausea	1	Other
			Pruritus	1	Other
			Rash	1	Other
		Spikevax	Dizziness	1	Threatening
			Gait disturbance	1	Threatening
			Hyperpyrexia	1	Other
			Hypotension	1	Threatening
			Loss of consciousness	1	Threatening
			Vision blurred	1	Threatening
Second dose	Matched controls	Vaxzevria	Respiratory arrest	1	Other
			Breast cancer	1	Other
			Myocardial infarction	1	Other
		Comirnaty	Hypertension	1	Hospital
			Malaise	1	Hospital
		Spikevax	Abortion, spontaneous	2	Hospital
	Prior SARS-CoV-2 infection	Vaxzevria	Retinal detachment	1	Hospital
			Vitreous floaters	1	Hospital
		Comirnaty	Depression	1	Other
			Eye hemorrhage	1	Other
			Fatigue	1	Other
			Headache	1	Other
			Nausea	1	Other
		Spikevax	Dizziness	1	Threatening
			Gait disturbance	1	Threatening
			Hypotension	1	Threatening
			Loss of consciousness	1	Threatening
			Vision blurred	1	Threatening
		Comirnaty	Hypothermia	1	Other
Booster dose	Matched controls	Comirnaty	Tachycardia	1	Other
			Congenital anomaly	1	Other
	Prior SARS-CoV-2 infection	Comirnaty	Hyperpyrexia	1	Threatening

Abbreviations: ADR = adverse drug reaction; CIOMS = The Council for International Organizations and Medical Sciences.

## Data Availability

Data are contained within the article and Appendix A.

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
