# Peer review of "Safety Monitoring of COVID-19 Vaccines in Persons with Prior SARS-CoV-2 Infection: A European Multi-Country Study"

_vaccines, 2024, doi:10.3390/vaccines12030241_

Round 1
Reviewer 1 Report
Comments and Suggestions for Authors
Although well designed and written, this study is currently of limited relevance, since both the circulating virus and repeated vaccinations have significantly reduced the number of non-immune subjects.
It would be much more interesting, in my opinion, to evaluate the percentage and severity of adverse events occurring in subjects suffering from particular pathologies, which can predispose to even fatal reactions (e.g. hemorrhages, thrombocytopenia, paroxysmal hypertension, ischemia, etc.), in various age groups and carry out a comparative analysis between those vaccinated with a previous full-blown disease and those who have never tested positive for the virus. This study could help doctors for a preventive evaluation of the risk-benefit ratio for the different categories of subjects. Furthermore, this analysis would also be useful in order to further inform the population on the lower extent of the risks related to vaccination compared to those related to viral infectionComments on the Quality of English Language
Acceptable quality of scientific English used by the Authors
Author Response
Comment: Although well designed and written, this study is currently of limited relevance, since both the circulating virus and repeated vaccinations have significantly reduced the number of non-immune subjects.
Response: We thank the Reviewer for this comment. Pivotal trials excluded subjects with prior SARS-CoV-2 infection, so it was necessary to generate evidence on COVID-19 vaccines safety in this cohort. For this reason, the EMA funded the “Covid Vaccine Monitor” project, with a focus on special cohorts of vaccinees including people with history of SARS-CoV-2 infection. During the whole study period we shared periodic safety reports with the EMA. We are conscious that repeated vaccinations and the rapid spread of the virus reduced the number of non-immune subjects; however, we do believe that this paper adds useful information in general on the effects of COVID-19 vaccines which may increase current knowledge about COVID-19 infection and vaccines as well as for other pandemics.
We also provided an overview of the safety of COVID-19 vaccines, across different vaccine brands, doses and countries, also comparing findings with those from the general population (recruited in the same study).
Comment: It would be much more interesting, in my opinion, to evaluate the percentage and severity of adverse events occurring in subjects suffering from particular pathologies, which can predispose to even fatal reactions (e.g. hemorrhages, thrombocytopenia, paroxysmal hypertension, ischemia, etc.), in various age groups and carry out a comparative analysis between those vaccinated with a previous full-blown disease and those who have never tested positive for the virus. This study could help doctors for a preventive evaluation of the risk-benefit ratio for the different categories of subjects. Furthermore, this analysis would also be useful in order to further inform the population on the lower extent of the risks related to vaccination compared to those related to viral infection
Response: We thank the Reviewer for the suggestions. No fatal outcome was observed and the results showed a low rate of serious adverse reactions. As for the proposed analysis, it is certainly of great relevance. Based on the study design and structure of our database, to investigate a potential interaction effect of medical history on the association between previous SARS-CoV-2 infection and the occurrence of at least one ADR, logistic regression models were performed. The estimated interaction effect was represented in a forest plot (Supplementary Figure 3). We updated the methods (row 183), the results (row 273), and the discussion section (row 405) accordingly.
Reviewer 2 Report
Comments and Suggestions for Authors
Thank you for giving me the opportunity to review this really interesting and relevant piece of work. This study present a very robust evidence of safety of SARS-CoV-2 vaccines either at first dose and after a booster dose in individuals with or without previous infection. Results are according to evidence published to date with study less robust designs, which reassures the quality and reliability of the results. Limitations are well identified and discussed.
However, I have just a couple of comments which might improve some secondary aspects of the research:
- In my understanding the important lost-to-follow up during the different Q1 to Q6 questionnaires which has been well noted by the authors, has not been fully discussed in terms of the biases introduced. I would appreciate if the authors could extend more about this and hypothesize with more detail in which direction this might biaised the results. They should include a more detailed cross analysis (maybe as a supplementary material) between the patients LTFU in different branches to ascertain if this have introduced any change in the participants profile and therefore the comparability of the outcomes observed.
- Secondly, I'm not sure if the time since the SARS-CoV-2 infection occurred have been recorded and included in the analysis. It is well known the waning of antibodies over time, which may in turn influence the occurrence of ADR. If not, introduce a sentence how this could have biased the results observed.
-Related to the point above, different SARS-CoV-2 strains have been overlapping since the beginning of the pandemic: from the ancestral strain, to the Beta, Delta and Omicron variants and subvariants. A major change in immune response and clinical presentation was observed specially between the transition from Delta to Omicron, and during the Omicron period it is expected that more unspecific or unnoticed infections may have occurred. The exposition to an specific strain among those with a prior infection might have affected the occurrence of ADR, or even the recall of an infection (for instance, more people might recall a Delta/Beta infection compared to those infected during the Omicron period). These limitations could be overcome including the time from infection, if available, or discussed in the limitations section.
In any way, I congratulate the authors for this important piece of work and I recommend their publication.
Author Response
Comment: In my understanding the important lost-to-follow up during the different Q1 to Q6 questionnaires which has been well noted by the authors, has not been fully discussed in terms of the biases introduced. I would appreciate if the authors could extend more about this and hypothesize with more detail in which direction this might biaised the results. They should include a more detailed cross analysis (maybe as a supplementary material) between the patients LTFU in different branches to ascertain if this have introduced any change in the participants profile and therefore the comparability of the outcomes observed.
Response: We thank the Reviewer for this comment. Following the first vaccination cycle, there has been a growing awareness of adverse events associated with the administration of COVID-19 vaccines. This may have made individuals less motivated to continue filling out the follow-up questionnaires, contributing to loss to follow-up. We added a sentence in the limitations section (row: 433), as follows: “Furthermore, after the first cycle of vaccinations, there was an increasing consciousness about the safety of these vaccines, which may have made vaccinees less motivated to continue filling out the FUQs contributing to loss to follow-up.”
We also report in the discussions in this article that, in line with pivotal trials and other observational studies in the literature, most ADRs are reported in the first 8 days corresponding to our initial follow-up questionnaires (Q1 and Q2).
Furthermore, loss to follow-up will be widely explored by an ad-hoc article focusing the LTFU on data from the Covid Vaccine Monitor project which will be available in the coming months. For this reason, we decided not to overlap the contents of the various articles that will be produced by the Covid Vaccine Monitor network.
Comment: Secondly, I'm not sure if the time since the SARS-CoV-2 infection occurred have been recorded and included in the analysis. It is well known the waning of antibodies over time, which may in turn influence the occurrence of ADR. If not, introduce a sentence how this could have biased the results observed.
Response: We thank the reviewer for pointing this out. We agree with this comment. Some participants reported the date of onset of symptoms on the web-app. To explore this topic, an analysis was conducted by stratifying people with previous SARS-CoV-2 infection in 4 different time intervals between the vaccination date and the start of symptoms although the analysis is underpowered. The results of this analysis are provided in Supplementary Table 7 and in the results section.
Comment: Related to the point above, different SARS-CoV-2 strains have been overlapping since the beginning of the pandemic: from the ancestral strain, to the Beta, Delta and Omicron variants and subvariants. A major change in immune response and clinical presentation was observed specially between the transition from Delta to Omicron, and during the Omicron period it is expected that more unspecific or unnoticed infections may have occurred. The exposition to a specific strain among those with a prior infection might have affected the occurrence of ADR, or even the recall of an infection (for instance, more people might recall a Delta/Beta infection compared to those infected during the Omicron period). These limitations could be overcome including the time from infection, if available, or discussed in the limitations section.
Response: We thank the Reviewer for this suggestion. In our study we did not collect biological samples to be able to distinguish the viral strain.
However, to explore this topic, we conducted an additional analysis using the WHO periodic bulletin on the prevalence of SARS-CoV-2 variants. Two periods, Alpha and Delta, were identified and participants were distributed between the two periods according to the date of onset of symptoms they had reported in the web app. The frequency of at least one ADR for each group period was calculated, and then the p-value was calculated to compare each group. We updated the methods (row: 186), the results (row: 302), and the limitations (row:440), accordingly. Supplementary table 8 provided the frequency of people with prior SARS-CoV-2 infection reporting at least one ADR, stratified by different SARS-CoV-2 variants of concern and dose.
Reviewer 3 Report
Comments and Suggestions for Authors
Manuscript by Ciccimarra et al, aimed at providing an over view of the ADRs after COVID-19 vaccines in particular for people with prior SARS-CoV2 infection. I find this topic of high importance but I feel that the manuscript would benefit from additional modifications before being further considered for publication in the Vaccines journal. I provide recommendation in the following text.
Title: please modify “vaccine” to vaccines and “SARS-COV-2” to SARS-CoV-2
Abstract lacks conclusion and recommendations which need to underline the importance of these findings.
Introduction: across the manuscript please correct SARS-COV-2 to SARS-CoV-2
Sentence “vulnerable populations/higher risk-people, such as those with prior SARS-CoV-2 infection,” in particular those with prior infection are not necessarily vulnerable/higher risk groups, this needs to be specified and/or corrected and needs reference.
Some examples of recorded ADRs should be listed and elaborated in the introduction in order to give specific importance to the topic.
Please make distinction between the EU and Europe (more countries and diverse COVID-19 vaccines available), since here you clearly talk about EU countries and EMA as administrative body of the EU.
Methods: as prospective cohort it remains unclear for how long the study lasted and those finally recruited in February 2023 for how long were followed? This needs to be specified.
How are those with prior SARS-CoV-2 recorded? Was this also self-reported data? this needs to be specified. What about those with reinfections - were they excluded?
Results: Stratification based on the severity of SARS-CoV-2 infection would also be useful to have presented, or at least mentioned in the discussion by hypothesizing what might change.
Stratification by country or ethnicity? Would it be useful?
Comparison between time to onset and time to recovery before vaccine presented in Fig6A lacks statistical analysis to understand the extent of difference?
Discussion needs to be better structured and would need to hypothesis why findings are as presented and not just to simply confront with what is in the literature.
Major limitation is that this is self-selected participation of those with prior SARS-CoV-2 and self-evaluated for ADR, and this needs to be clearly stated in the limitations. Also this might underestimated the true prevalence of ADR in the population(s). Also, if there was no data on clinical severity of infection (since was not presented before), this needs to be clearly specified in limitations.
Conclusion needs to be better structured, it practically lack any finding(s) from this study. First two sentences are a general statement not necessarily coming from these results.
Author Response
Comment: Title: please modify “vaccine” to vaccines and “SARS-COV-2” to SARS-CoV-2
Response: We thank the Reviewer for this comment. We edited the text accordingly.
Comment: Abstract lacks conclusion and recommendations which need to underline the importance of these findings.
Response: We thank the Reviewer for this comment. We edited the text accordingly.
Comment: Introduction: across the manuscript please correct SARS-COV-2 to SARS-CoV-2
Response: We thank the Reviewer for this comment. We edited the text accordingly.
Comment: Sentence “vulnerable populations/higher risk-people, such as those with prior SARS-CoV-2 infection,” in particular those with prior infection are not necessarily vulnerable/higher risk groups, this needs to be specified and/or corrected and needs reference.
Response: We thank the Reviewer for this comment. We edited the text as suggested and added a reference. Please see row: 66
Comment: Some examples of recorded ADRs should be listed and elaborated in the introduction in order to give specific importance to the topic.
Response: We thank the Reviewer for this comment. We added a sentence about ADRs in pivotal trials in the introduction section. Please see row: 61
Comment: Please make distinction between the EU and Europe (more countries and diverse COVID-19 vaccines available), since here you clearly talk about EU countries and EMA as administrative body of the EU.
Response: We thank the reviewer for this comment. As suggested, we specified that we monitored the safety of EMA-approved COVID-19 vaccines (at the time of the study) in the 11 European countries that participated in the project.
Comment: Methods: as prospective cohort it remains unclear for how long the study lasted and those finally recruited in February 2023 for how long were followed? This needs to be specified.
Response: We thank the reviewer for this comment. We updated the methods section accordingly. Please see row: 147
Comment: How are those with prior SARS-CoV-2 recorded? Was this also self-reported data? this needs to be specified. What about those with reinfections - were they excluded?
Response: We thank the reviewer for this comment. Yes, also this information was self-reported and no clinical assessment was performed. We updated the limitations section accordingly (row: 443)
The cohort prior SARS-CoV-2 infection includes all subjects who reported having had a previous infection at the time of study registration, irrespective of any reinfection during the follow-up period.
Comment: Results: Stratification based on the severity of SARS-CoV-2 infection would also be useful to have presented, or at least mentioned in the discussion by hypothesizing what might change.
Response: We thank the reviewer for this suggestion. The severity of SARS-CoV-2 infection was evaluated based on the symptoms reported by the vaccinees. To explore this topic, we analyzed the frequency of at least one ADR in people with prior SARS-CoV-2 infection, stratified by different criteria of symptoms severity. We updated the methods section (row: 190) and the results (row: 305) accordingly. The results of this analysis are provided in Supplementary Table 9.
Comment: Stratification by country or ethnicity? Would it be useful?
Response: We thank the reviewer for this comment. Regarding stratification by country, Table 10 was added in the supplementary materials, while ethnicity could not be reported in the web app, so a sentence was added in the limitations sections. (row: 442)
Comment: Comparison between time to onset and time to recovery before vaccine presented in Fig6A lacks statistical analysis to understand the extent of difference?
Response: We thank the reviewer for this comment. A Wilcoxon test was performed to compare TTO and TTR. We updated the methods section (row: 211) and the results (row: 336) accordingly.
Comment: Discussion needs to be better structured and would need to hypothesis why findings are as presented and not just to simply confront with what is in the literature.
Response: We thank the reviewer for this comment. We added some sentences to provided better explanation for some findings. Please see row: 383
Comment: Major limitation is that this is self-selected participation of those with prior SARS-CoV-2 and self-evaluated for ADR, and this needs to be clearly stated in the limitations. Also this might underestimated the true prevalence of ADR in the population(s). Also, if there was no data on clinical severity of infection (since was not presented before), this needs to be clearly specified in limitations.
Response: We thank the reviewer for this comment. We added these constructive criticisms in the limitations section. Please see row: 443
Comment: Conclusion needs to be better structured, it practically lack any finding(s) from this study. First two sentences are a general statement not necessarily coming from these results.
Response: We thank the reviewer for this comment. We updated the conclusion accordingly.
Reviewer 4 Report
Comments and Suggestions for Authors
In this manuscript, the authors report findings from a large-scale prospective study of COVID-19 vaccinees from eleven European countries. This web-based study using electronic questionnaires collected and statistically analyzed subject-reported outcomes related to the type and frequency of adverse drug reactions in patients receiving different COVID-19 vaccines (Vaxzevria, Comirnaty, Spikevax, and Jcovden) who had a previous SARS-CoV-2 infection. The experimental study design described is scientifically sound and appropriate for the study's specific aims. The results are clearly reported, and the conclusions are supported by the authors' findings, which add to the current literature on the safety of COVID-19 vaccines.
Comments on the Quality of English LanguageOverall, the quality of the English is acceptable. Some minor editing of the English in this manuscript could be carried out to improve the quality.
Author Response
We thank the Reviewer for the words of appreciation.
Round 2
Reviewer 3 Report
Comments and Suggestions for Authors
Thank you for addressing my concerns and providing respose to comments.
Comments on the Quality of English LanguageNo major issues